# H-Bonds, π-Stacking and (Water)O-H/π Interactions in (μ4-EDTA)Bis(Imidazole) Dicopper(II) Dihydrate

Jeannette Carolina Belmont-Sánchez [1], María Eugenia García-Rubiño [2], Antonio Frontera [3], Josefa María González-Pérez [1], Alfonso Castiñeiras [4] and Juan Niclós-Gutiérrez [1,*]

1   Department of Inorganic Chemistry, Faculty of Pharmacy, University of Granada, 18071 Granada, Spain; carol.bs.quimic@hotmail.com (J.C.B.-S.); jmgp@ugr.es (J.M.G.-P.)
2   Departamento Fisicoquímica, Facultad de Farmacia, Universidad de Granada, 18071 Granada, Spain; rubino@ugr.es
3   Departament de Química, Universitat de les Illes Balears, Crta. de Valldemossa km 7.5, 07122 Palma de Mallorca (Baleares), Spain; toni.frontera@uib.es
4   Department of Inorganic Chemistry, Faculty of Pharmacy, University of Santiago de Compostela, 15782 Santiago de Compostela, Spain; alfonso.castineiras@usc.es
*   Correspondence: jniclos@ugr.es

**Abstract:** We synthesized and studied the polymeric compound $\{[Cu_2(\mu_4\text{-}EDTA)(Him)_2]\,2H_2O\}_n$ (**1**). The single-crystal structure is reported along with an in depth characterization of its thermal stability (TGA), spectral properties (FT-IR, Vis-UV and RSE), and magnetic behavior. The crystal consists of infinite 2D-networks built by centrosymmetric dinuclear motifs, constructed by means of a bridging *anti,syn*-carboxylate group from each asymmetric unit. Each layer guides Him ligands toward their external faces. They are connected by intermolecular (Him)N-H···O(carboxylate) bonds and antiparallel π–π stacking between symmetry related pairs of Him ligands, and then pillared in a 3D-network with parallel channels, where disordered water molecules are guested. About half of the labile water is lost from these channels over a wide temperature range (r.t. to 210 °C) before the other one, most strongly retained by the cooperating action of (water)O1-H(1A)···O(carboxylate) and (water) O1-H(1B)···π(Him) interactions. The latter is lost when organic ligands start to burn. ESR spectra and magnetic measurements indicated that symmetry related Cu(II) centers connected by the bridging carboxylate groups behave magnetically not equivalently, enabling an exchange interaction larger than their individual Zeeman energies.

**Keywords:** crystal structure; copper(II); EDTA; imidazole; polymer; H-bonding; π–π stacking; water O-H/π interaction



## 1. Introduction

Ethylenediaminetetraacetic acid ($H_4$EDTA) and its different anionic forms generate an enormous diversity of metal chelate complexes. The ability of EDTA to form up to five stable five-membered chelate rings around a metal center is only one among all these possibilities, frequently enriched in structurally well documented compounds where μ-EDTA bridging forms increases its denticity. Now we are interested in using the tetravalent EDTA anion as a bridging μ-chelator for two metallic centers (M), thus generating $M_2$(μ-EDTA) motifs which can form dinuclear complex molecules, as well as polymeric networks of different dimensionalities. In these compounds, at least a tridentate role for each half-EDTA should be expected, where each M center should be chelated by a *N,N*-methylene-aminodiacetate(2-) group, $-CH_2\text{-}N(CH_2CO_2^-)_2$ (here after a mida group). A search in the Cambridge Structural Database (CSD, version 5.41, update August 2020) for "any metal" (M) in $M_2$(μ-EDTA) motifs affords 58 different structures (number of examples for each M indicated in parenthesis): Mo (36), W (6), Cu(5), U (3), Tc (2), Sn (2), Re (1), Zr (1), Al (1), or Ba (1). Since some literature is certainly old and/or is not in English, a reference code in

CSD database is provided for such compounds in the text. The highest $\mu_n$-EDTA denticity in these kind of compounds has been reported for the Ba(II) derivative, [Ba$_2$($\mu_{12}$-EDTA)] (TAHFUF in CSD) [1].

As far as concerns M$_2$($\mu$-EDTA) compounds (M being now a first-row transition metal ion) the crystal structures of salts [Fe$^{II}$(H$_2$O)$_6$][Mo$_2$($\mu$-EDTA)($\mu$-O)$_2$]·5H$_2$O (UCETAX) [2] and [Ni(H$_2$O)$_6$][Mo$_2$($\mu$-EDTA)($\mu$-S)$_2$]·2H$_2$O (LOVKUE) [3] have been reported along with those of five Cu(II) compounds with "Cu($\mu$-EDTA)Cu" motifs [4–7]. These copper(II) compounds comprise, among others (see below) two 2D-polymers also having Na(I) ions, Kegging poly-oxo-wolframate anions, as well as bounded to Na(I)-aqua ligands and crystallization water molecules [4]. The other three are {[Cu$_2$($\mu_4$-EDTA)(H$_2$O)$_2$]·2H$_2$O}$_n$ (CUEDTA01, here after the so-called the binary dinuclear chelate) [5], [Cu$_2$($\mu_4$-EDTA)(py)$_2$(H$_2$O)$_2$]·2H$_2$O (py = pyridine, NAMJOB) [6], and {[Cu$_2$($\mu_4$-EDTA)(3hpy)$_2$]}$_n$ (3hpy = 3-hydroxypyridine, PEZRES ) [7]. See additional comments on the three compounds in Sections 3.1 and 3.2

As a part of a program on ternary copper(II) complexes with amino-polycarboxylate chelators and five (imidazole) or six-membered (pyrimidine's) *N*-heterocyclic ligands related to structural formulas of purine nucleic basis, here we report the synthesis, crystal structure, (thermal, spectral and magnetic) physical properties, and density functional theory (DFT) calculations of {[Cu$_2$($\mu_4$-EDTA)(Him)$_2$]·2H$_2$O}$_n$ (**1**).

## 2. Materials and Methods

### 2.1. Reagents

Malachite Cu$_2$CO$_3$(OH)$_2$ (Aldrich), H$_4$EDTA acid (TCI) and imidazole (Alfa Aesar) were used as received.

### 2.2. Crystallography

A blue needle crystal of {[Cu$_2$($\mu_4$-EDTA)(Him)$_2$]·2H$_2$O}$_n$ (**1**) was mounted on a glass fiber and used for data collection. Diffraction data were obtained using a Bruker D8 VENTURE PHOTON III-14 diffractometer (Bruker AXS GmbH, Karlsruhe, Germany). The data were processed with APEX2 [8] and corrected for absorption using SADABS [9]. The structure was solved by direct methods using the program SHELXS-2013 [10], and refined by full-matrix least-squares techniques against F$^2$ using SHELXL-2013 [10]. The O1 water atom is disordered over two positions; the occupancy factor for each one was refined, resulting in a value of 0.440(8) for O1 and 0.560(8) for O2. Positional and anisotropic atomic displacement parameters were refined for all non-hydrogen atoms. Hydrogen atoms were located in difference maps and included as fixed contributions riding on attached atoms with isotropic thermal parameters constrained to 1.2/1.5 U$_{eq}$ of the carrier atoms. Molecular graphics were generated with DIAMOND [11]. Crystal data, experimental details, and refinement results are summarized in Table 1. Crystallographic data for 1 has been deposited in the Cambridge Crystallographic Data Center, with the CCDC number 2047832.

### 2.3. Other Physical Measurements

Analytical data (CHN) was obtained in a Fisons–Carlo Erba EA 1108 elemental microanalyzer (Carlo Erba Reagents, Sabadell, Spain). The copper(II) content was cheeked as CuO by the weight of the final residue in the thermo-gravimetric analysis (TGA) at 950 °C within <1% of assumed experimental error. The FT-IR spectrum was recorded (KBr pellet) on a Jasco FT-IR 6300 spectrophotometer (JASCO Deutschland GmbH, Pfungstadt, Germany). The electronic (diffuse reflectance) spectrum was obtained with a Varian-Cary5E spectrophotometer (Agilent Technologies, Inc., Santa Clara, CA, USA). TGA was carried out (r.t.–950 °C) in airflow (100 mL/min) by a Shimadzu Thermobalance TGA–DTG–50H (SHIMADZU DEUTSCHLAND GmbH, Duisburg, Germany), while a series of 36 time-spaced FT-IR spectra were recorded with a coupled FT-IR Nicolet Magna 550 spectrometer (Thermo Fisher Scientific Inc., Madrid, Spain) to identify the evolved gasses. X-band EPR measurements were carried out on a Bruker ELEXSYS 500 spectrometer (Karlsruhe,

Germany) equipped with a super-high-Q resonator ER-4123-SHQ and standard Oxford low temperature devices. For Q-band studies, EPR spectra were recorded on a Bruker EMX system equipped with an ER-510-QT resonator. An NMR probe calibrated the magnetic field, and the frequency inside the cavity was determined with a Hewlett-Packard 5352B microwave frequency counter (Agilent Technologies, Madrid, Spain). Computer simulation: WINEPR-SimFonia, version 1.5, Bruker Analytische Messtechnik GmbH (Karlsruhe, Germany). Temperature dependent magnetic measurements were performed between 2 and 300 K with an applied field of 0.1 T using a commercial MPMS3 SQUID magnetometer (Quantum Design GmbH, Darmstadt, Germany). The experimental susceptibilities were corrected for the diamagnetism of the constituent atoms using Pascal tables.

**Table 1.** Crystal data and structure refinement for $\{[Cu_2(\mu_4\text{-EDTA})(Him)_2]\cdot 2H_2O\}_n$ (**1**).

| **Empirical Formula** | $C_8H_{12}CuN_3O_5$ |
|---|---|
| Empirical formula weight | 293.75 |
| Temperature | 100(2) K |
| Wavelength | 0.71073 Å |
| Crystal system, space group | Monoclinic, C2/c |
| Unit cell dimensions | a = 21.1313(13) Å, $\alpha$ = 90° |
| | b = 5.9280(4) Å, $\beta$ = 110.474(3)° |
| | c = 18.1842(13) Å, $\gamma$ = 70.545(2)° |
| Volume | 2134.0(3) Å$^3$ |
| Z, Calculated density | 8, 1.829 Mg/m$^3$ |
| Absorption coefficient | 2.062 mm$^{-1}$ |
| F(000) | 1200 |
| Crystal size | $0.250 \times 0.070 \times 0.040$ mm$^2$ |
| Theta range for data collection | 3.588 to 30.505$^0$ ° |
| Limiting indices | $-28 \leq h \leq 30, -8 \leq k \leq 8, -25 \leq l \leq 25$ |
| Reflections collected / unique | 32813 / 3255 [R(int) = 0.0462] |
| Completeness to $\theta$ = 25.242 | 99.7% |
| Absorption correction | Semi-empirical from equivalents |
| Max. and min. transmission | 1.000 and 0.857 |
| Refinement method | Full-matrix least-squares on F$^2$ |
| Data/restraints/parameters | 6551/0/307 |
| Goodness-of-fit on F$^2$ | 1.055 |
| Final R indices [I > 2σ(I)] | $R_1$ = 0.0278, w$R_2$ = 0.0619 |
| R indices (all data) | $R_1$ = 0.0354, w$R_2$ = 0.0663 |
| Largest diff. peak and hole | 0.461 and $-$ 0.590 e.Å$^{-3}$ |
| CCSD ref. number | 2047832 |

### 2.4. Synthesis with Relevant Vis-UV and FTIR Spectral Data

Compound **1** was obtained in a two-step process. First, $Cu_2CO_3(OH)_2$ (green malachite, 1 mmol, 0.22 g) and $H_4$EDTA (1 mmol, 0.29 g) (Alpha-Aesar, Kandel, Germany) were reacted in water (100 mL) inside an open Kitasato flask at 50 °C. Magnetic stirring was aided by hand shaking from time to time to promote a full reaction of these products, to give a clear blue solution of $Cu_2(\mu\text{-EDTA})$ chelate. Heat was then ceased, and the solution filtered (without vacuum) to remove any malachite residue. Him (2 mmol, 0.19 g) was added to the filtrate, which produced an intensification of the blue color in the resulting solution. Slow evaporation of this mother liquor (about one week at r.t.) produced the first crystals of **1** which were collected by filtration, washed with water, and air-dried to be checked by FT-IR spectroscopy. The remaining mother liquor was placed in a crystallizer, and put inside a desiccator to permit a slow diethyl ether diffusion. The formation of blue crystals of **1** (many of them suitable for crystallographic purposes) was achieved in a high yield (~0.50 g, ~85%). Elemental analysis (%): Calc. for $C_{16}H_{24}Cu_2N_6O_{10}$: C 32.71, H 4.12, N 14.31, Cu (as CuO) 27.08; Found: C 32.67, H 4.10, N 14.29, Cu 27.09 (as CuO, final residue at 950 °C, in the TGA curve). UV–vis spectrum data: rather symmetrical d-d band with maximum absorption at 665 nm. FT–IR data (cm$^{-1}$): 3650–3200vbr $\nu_{as}/\nu_s(H_2O)$,

peaks at 3161w, 3144w, 3124w $\nu_{as}$(NH), 3045w $\nu$(C-H)$_{arom}$, 2988, 2934 $\nu_{as}$(CH$_2$), 2866, 2814 $\nu_{as}$(CH$_2$), ~1640sh $\delta$(H$_2$O), 1611 $\nu_{as}$(COO), 1550vw or 1512vw $\delta$(N-H), 1389m $\nu_s$(COO), 910w or 902w $\pi$(C-H)$_{arom}$ for only one C-H (usually expected at 900-860), 840w $\pi$(C-H)$_{arom}$ for two adjacent C-H (expected at 860-810). These spectra are reported as Supporting Information Figures S1 and S2, respectively.

### 2.5. Theoretical Methods

The energies of the assemblies analyzed in this manuscript were computed using the Gaussian-16 program [12], as follows. The interaction energies $\Delta E$, were computed as the energy difference between the dimeric assembly and the sum of the energies of the monomers. The basis set superposition error was corrected using the Boys and Bernardi approach [13]. The def2-TZVP [14,15] basis set combined with the PBE0 functional [16,17] and the D3 dispersion correction [18,19] were used for the calculations. This level of theory (functional and basis set) has been used before to study noncovalent interactions in the solid state [20–24], including those analyzed in this work [25–27]. Moreover, X-ray coordinates instead of optimized geometries were used to investigate the energetic features of the noncovalent interactions in the solid state, because we were interested to analyze them as they stand in the solid state, instead of investigating the optimal complexation geometry. The MEP (molecular electrostatic potential) surfaces were computed using the 0.001 isosurface and visualized using Gaussview software [12] at the same level of theory. The noncovalent interaction (NCI) index [28,29] and quantum theory of atoms-in-molecules (QTAIM) [30,31] analyses were calculated using the wavefunction generated by Gaussian-16 as input for the AIMAll program [32].

## 3. Results and Discussion

### 3.1. A Comment About the Synthesis of Compound **1**

The two step synthetic procedure here reported for **1** agrees with the desired stoichiometric ratio Cu:EDTA:Him 2:1:2. If Him is not added (second step), the reaction between malachite and H$_4$EDTA yields the binary dinuclear chelate, {[Cu$_2$($\mu_4$-EDTA)(H$_2$O)$_2$]·2H$_2$O}$_n$, whose crystal structure has been previously reported [5]. On the other hand, Sergienko et al. [6], reacting this binary dinuclear chelate (as stating reactant) and a large excess of Him (Him:Cu mole ratio = ~7:1) synthesized and X-ray characterized the salt [Cu(Him)$_6$][(Him)($\mu_2$-EDTA)Cu·Cu(Him)$_4$·Cu($\mu_2$-EDTA)(Him)]·6H$_2$O (GEMPOE in CSD, with an averaged Him/Cu mole ratio 3/1) [6]. In this compound the centrosymmetric trinuclear complex anion has EDTA as pentadentate chelator (having a non-coordinated acetate arm) for both the terminal Cu(II) atoms as well as the $\mu_2$-bridging ligand for the central Cu(Him)$_4$ linker moiety. This enables the N-binding of Him in one of the four closest sites of the elongated coordination of the metal center at both terminal Cu($\mu_2$-EDTA)(Him) moieties.

### 3.2. Thermal Stability

Under air-dry flow, the weight loss versus temperature TGA behavior of **1** consists of six steps (Figure 1). Experimental results, calculated loss weights, and evolved gases or final residue are reported in Table 2. Detailed information of this TGA is supplied as Supporting Information Figure S3.

First, compound **1** loses one of two uncoordinated water molecules (calculated for 2 H$_2$O molecules 6.133%!) before starting the burning of organic ligands above 210 °C. Thus, the remaining water at 210 °C will be most likely lost during the second step, overlapped with the beginning of burning of organic ligands. The evolved gases suggest that this phenomenon first affects EDTA, but weight loss in the second to sixth steps advise against additional speculative attributions. Even so, it seems evident that the aromatic Him ligands burn over 500 °C. Most organics burn below 515 °C, yielding a non-pure CuO residue. Interestingly, a small weight loss during the last step leads to a final residue in excellent agreement with the calculated value for a pure CuO residue.

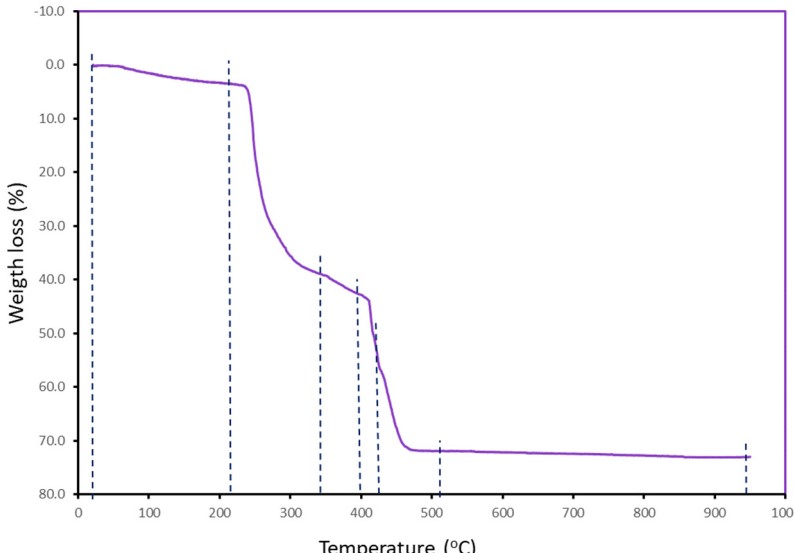

**Figure 1.** Weight loss versus temperature (in the range r.t. to 950 °C) in the thermo-gravimetric analysis (TGA) analysis of **1** (sample: 11.03 mg).

**Table 2.** Summary of results obtained from the thermogravimetric analysis of compound **1**.

| Step or R | Temp. (°C) | Time (min) | Weight (%) Exp. | Cal. | Evolved Gases or Residue (R) |
|---|---|---|---|---|---|
| 1 | 40–210 | 2–19 | 3.366 | 3.066 * | 1 $H_2O$ *, $CO_2$ (t) |
| 2 | 210–345 | 19–33 | 33.474 | - | $H_2O$, $CO_2$, CO (t) |
| 3 | 345–400 | 33–38 | 3.704 | - | $H_2O$, $CO_2$, CO, $CH_4$ (t) |
| 4 | 400–425 | 38–42 | 14.307 | - | $H_2O$, $CO_2$, CO, $CH_4$ (t) |
| 5 | 425–515 | 42–51 | 14.817 | - | $H_2O$, $CO_2$, CO, $CH_4$ (t) $N_2O$, NO, $NO_2$ |
| 6 | 515–950 | 51–91 | 1.317 | - | $H_2O$, $CO_2$, CO (t), $CH_4$ (t), $N_2O$, NO, $NO_2$ |
| R | 950 | 95 | 27.088 | 27.080 | 2 CuO |

* Calculated for the loss of only 1 $H_2O$. t = trace amounts.

### 3.3. Copper(II) Coordination and Crystal Structure

Compound **1** is a polymer that agrees with the formula {[$Cu_2(\mu_4$-EDTA)(Him)$_2$]·2$H_2O$}$_n$. It consists of centro-symmetric dinucelar complex motifs and non-coordinated water (Figure 2). In each half of the dinuclear complex motif, the Cu(II) center exhibits a distorted square-based pyramidal surrounding (Table 3), type 4+1. The square base is supplied by the N(1), O(11), and O(21) donors from a *mer*-mida group from EDTA and the N(1) imidazole donor, occupying the four shortest coordination sites of the Cu(II). One of the acetate arms of the mida group acts as O(21)-monodentate, whereas the other one displays an anti,syn-O(11),O(12)-bridging function. This $\mu_2$-carboxylate group represents the main driving promoter for the polymerization of **1**, supplying to each Cu(II) center of its distal O12$\neq$1 donor atom, at the largest bond distance of 2.488(1) Å. The square base of the coordination polyhedron is slightly distorted, and it reveals the Addison–Reedijk parameter (also the so-called trigonal index) τ, estimated for **1** from values of their trans-basal coordination angles (θ or φ) as τ = (θ−φ)/60 = (172.70−168.16)/60 = 0.076 (Table 3). All structural coordination parameters in **1** are in agreement with an expectable Jahn–Teller distortion for copper(II) complexes, always related to its stable electronic configurations ([Ar]3$d^9$) and the useful "hole formalism" (versus a 3$d^1$). As reported for the closely related compounds with pyridine coligands, [$Cu_2(\mu_4$-EDTA)(py)$_2$($H_2O$)$_2$]·2$H_2O$ [6] and {[$Cu_2(\mu_4$-EDTA)(3hpy)$_2$]}$_n$ [7], the efficient coordination of imidazole is due to the mutual affinity between the Pearson's borderline acid Cu(II) and N-heterocyclic borderline basis,

as well as to the *mer*-mida tridentate group. This enables the binding of the N-Him donor at one among the four closest donor sites around the copper(II) center. This is seen in many other related five-coordinated Cu(II) compounds (of the type 4+1) that also promote a displacement of the metal center (from the mean basal coordination plane) toward the distal donor atom (0.03 Å in **1**).

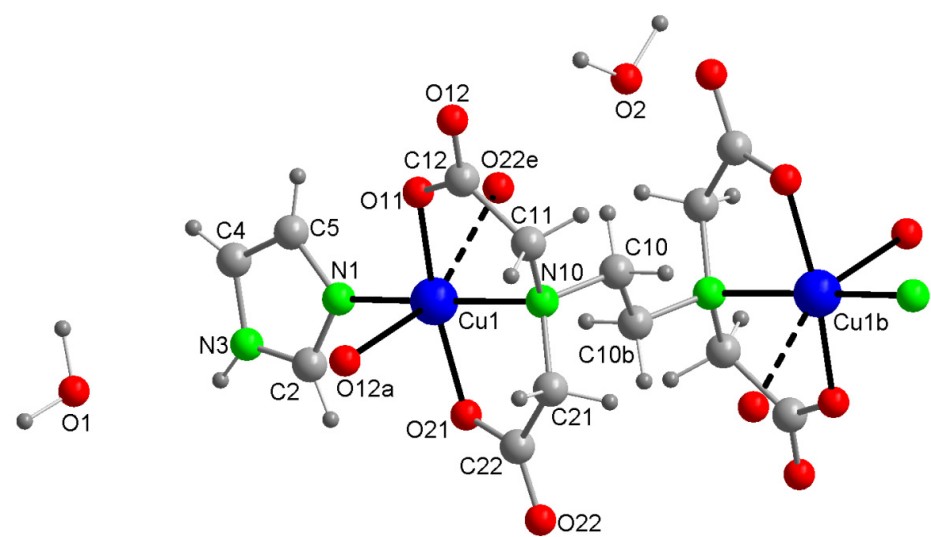

**Figure 2.** Structure and numbering scheme for non-hydrogen atoms in the asymmetric unit of **1**. The water molecule has partial occupancies in the two disordered represented positions. Symmetry codes: a = #1 = −x + 1/2, y − 1/2, −z + 1/2, b = #2 = −x + 1/2, −y + 1/2, −z, e = #5 = −x + 1/2, −y − 1/2, −z.

**Table 3.** Coordination bond lengths (Å) and angles (°) in the crystal of compound **1**, {[Cu$_2$($\mu_4$-EDTA)(Him)$_2$]·2H$_2$O}$_n$. See Figure 2 for numbering scheme of relevant atoms.

| Atoms | Distance or Angle |
|---|---|
| Cu(1)-O(11) | 1.9389(12) |
| Cu(1)-N(1) | 1.9451(14) |
| Cu(1)-O(21) | 1.9608(12) |
| Cu(1)-N(10) | 2.0207(14) |
| Cu(1)-O(12)#1 | 2.4878(12) |
| O(11)-Cu(1)-N(1) | 97.23(5) |
| O(11)-Cu(1)-O(21) | 168.16(5) ($\varphi$) |
| N(1)-Cu(1)-O(21) | 94.22(5) |
| O(11)-Cu(1)-N(10) | 84.79(5) |
| N(1)-Cu(1)-N(10) | 172.70(5) ($\theta$) |
| O(21)-Cu(1)-N(10) | 84.23(5) |
| O(11)-Cu(1)-O(12)≠1 | 88.90(5) |
| N(1)-Cu(1)-O(12)# 1 | 89.50(5) |
| O(21)-Cu(1)-O(12)#1 | 88.19(5) |
| N(10)-Cu(1)-O(12)#1 | 97.57(5) |

Symmetry code: #1 − x + 1/2, y − 1/2, −z + 1/2.

The crystal of **1** consists of an infinite network, having non-coordinated water molecules and polymeric 2D-layers of the metal complex. Such layers run parallel to the bc plane, in such a manner that Him ligands are oriented toward their external faces. They are then pillared along the a axis, connected by an extensive network of N(3)-H(3)···O(22) ≠ 3 interactions (Figure 3a and Table 4). Such interactions are efficiently reinforced by π-staking between appropriate pairs of symmetry related anti-parallel Him ligands from adjacent layers: inter-centroids distance d$_{c-c}$ = 3.45 Å (depicted in Figure 3b), inter-planar distance

$d_{\pi\text{-}\pi}$ = 3.24 Å, dihedral angle α = 0°, slipping angles β = γ = 20.2°). Consequently, both intermolecular interactions infer a strong 3D cohesion, at the same time generating funnels parallel to the *a* axis (Figure 3a) where water molecules are hosted with certain disorder, in two non-equivalent positions (see Section 2.2), identified by the O-number of water in Figure 2.

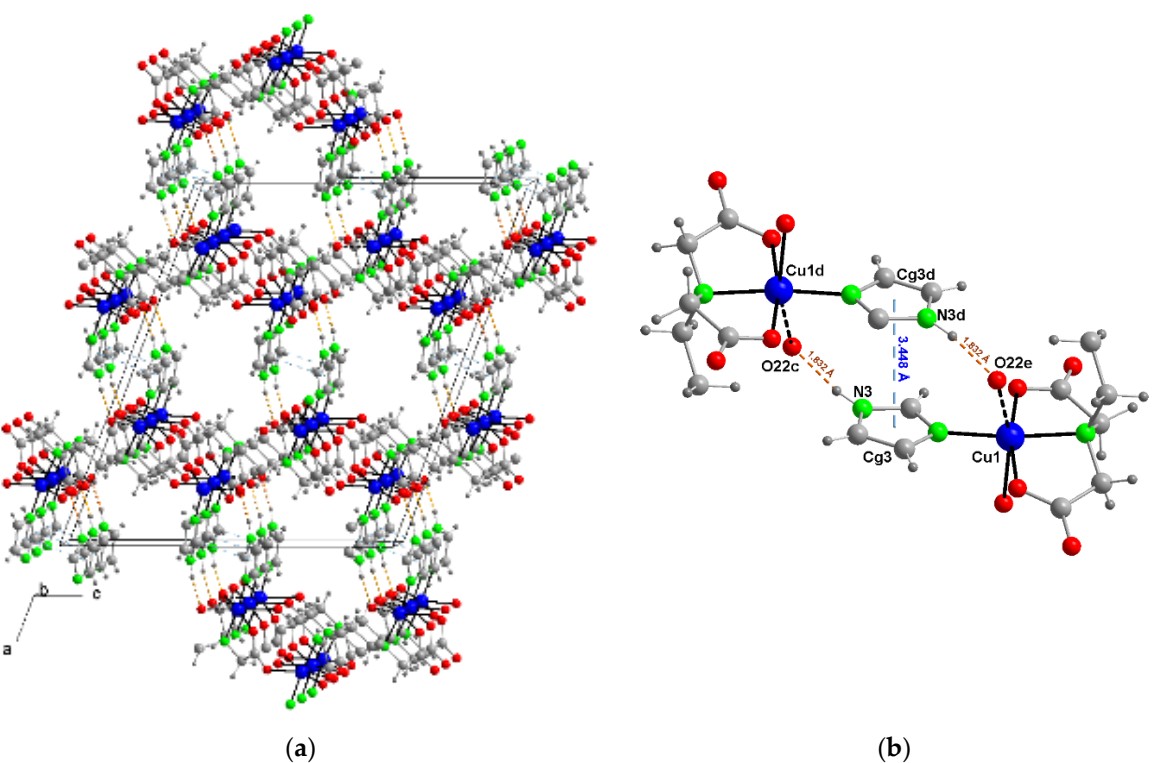

(**a**)             (**b**)

**Figure 3.** (**a**) Layers of the polymeric metal complex of **1**, pillared along the a axis of the crystal connected by (Him)N···O(carboxylate). Non-coordinated water and other intermolecular interactions omitted for clarity, to depict void channels where water is hosted. (**b**) Crystal structure fragment showing the π-stacking interaction between a pair of anti-parallel Him ligands, belonging to the external faces of two pillared layers of the polymeric metal complex of **1**.

**Table 4.** Hydrogen bonds in the crystal of $\{[Cu_2(\mu_4\text{-EDTA})(Him)_2]\cdot 2H_2O\}_n$ (**1**) [Å, °]. D = H-donor atom, A = H-acceptor atom.

| D-H···A | D(D···A) | <(D-H···A) |
|---|---|---|
| N(3)-H(3)···O(22) ≠ 3 | 2.774(2) | 169.8 |
| O(1)-H1A···O(12) ≠ 7 | 2.869(4) | 154.7 |

Symmetry codes: c = #3 = −x, −y, −z, d = ≠4 = −x + 1/2, y + 1/2, −z + 1/2, e = #5 = −x + 1/2, −y − 1/2, −z, g = #7 = x − 1/2, y − 1/2, z.

The observed "host" role for the 3D-crystal network here reported as "guest disordered water" requires some comments. First of all, the labile $O(2)H_2$ falls inside channels without being involved in identified intermolecular interactions. In clear contrast, the water numbered as $O(1)H_2$ is retained inside the channels by the cooperating action of two weak intermolecular forces: the O1-H1A···O12)≠7 H-bond (#7 = x − 1/2, y − 1/2, z, see Table 4) and the quite unusual O1-H1B/π(Him) interaction (Figure 4). Now, the TGA behavior of **1** can be rationalized, emphasizing that the "more labile water" is retained in a surprisingly broad range of temperatures (r.t.–210 °C), because it is hosted within channels. Starting from 210 °C, the hardly retained water should be lost, overlapped to the beginning of the combustion of organic ligands.

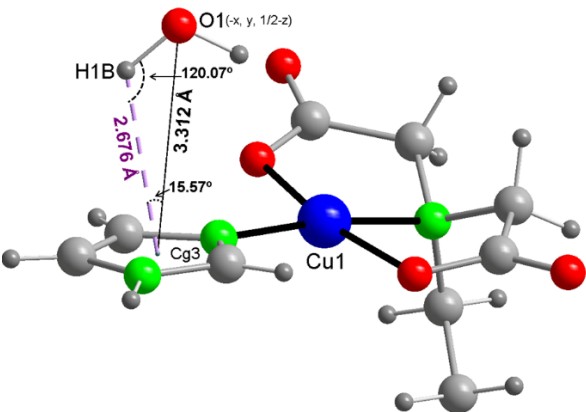

**Figure 4.** Crystal structure fragment representing the intermolecular (water)O1-H1B···π(Him) interaction. In brackets, assumable geometric parameters (d, Θ, Φ) typically used to recognize this kind of interaction [33–38]: d(O)···Him(Cg, centroid) 3.31 (4.3) Å, θ(O1-Cg-H1B) 15.57° (<25°), φ(O1-H1B···Cg) 120.1 (120–180°).

A graphical summary of all coordination bonds and intermolecular interactions contributing to the stability of **1** is illustrated in Figure 5.

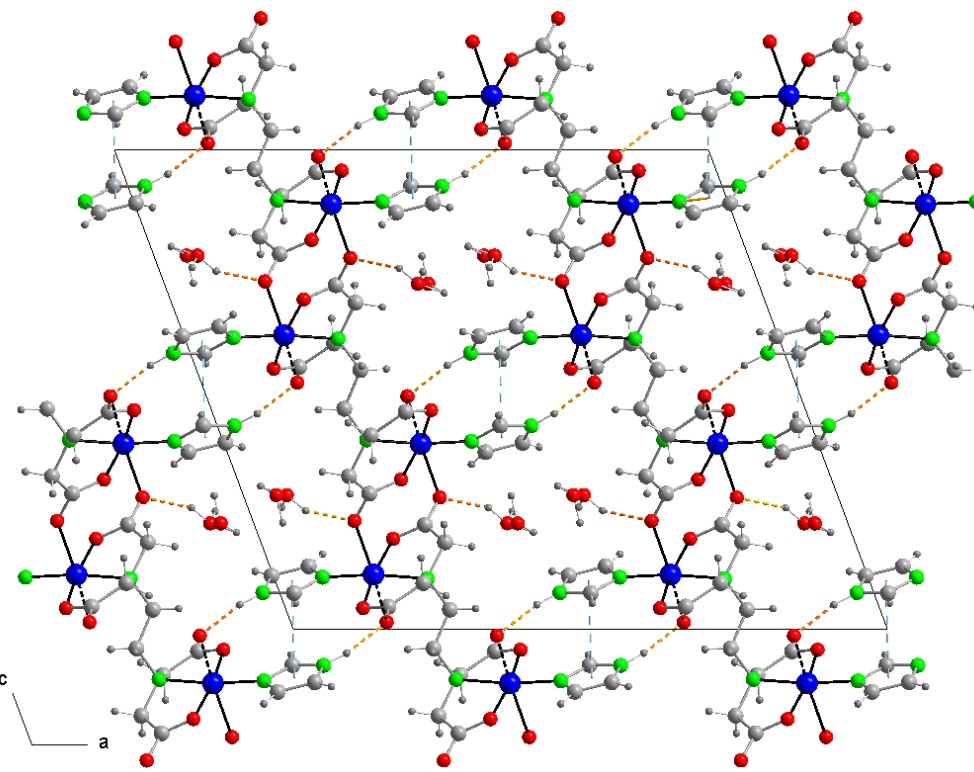

**Figure 5.** Intermolecular interactions contributing to the stability of the crystal of **1**, uniting the association of the 2D layered coordination polymer (N-H···O, π–staking) and reinforcing the retention of half the water (cooperation of O-H···O, O-H/π).

### 3.4. Magnetic Porperties

The X-band (9.395 GHz) EPR spectrum shows the characteristic features of a rhombic g tensor from 5 K to room temperature (Figure 6). The main components of the g tensor, estimated by the comparison of the experimental spectra with those obtained by a computer simulation program working at the second order of the perturbation theory, are $g_1$ = 2.194,

$g_2$ = 2.125, and $g_3$ = 2.066 (<g> = 2.128). It is notable that these parameters do not agree with the axially elongated square-base pyramidal coordination of copper in **1**. This fact suggests the presence of exchange interactions between magnetically non-equivalent $Cu^{2+}$ ions, as confirmed by the 2.0 value deduced for the Hathaway G parameter [36]. In this way, Cu1 and Cu1$^d$ (d = $\neq$4 = −x + 1/2, y + 1/2, −z + 1/2) related by a two-fold screw axis and connected by a anti,syn-carboxylate bridge, with Cu···Cu$^d$ separation of 5.766(1) Å) are crystallographically equivalents, but not from a magnetic point of view. If the exchange interaction (J) between them is larger than the difference between their Zeeman energies, the individual resonances must be averaged. This would explain why the observed spectrum does not adequately reflect the coordination geometry of the environment of the Cu(II) ions.

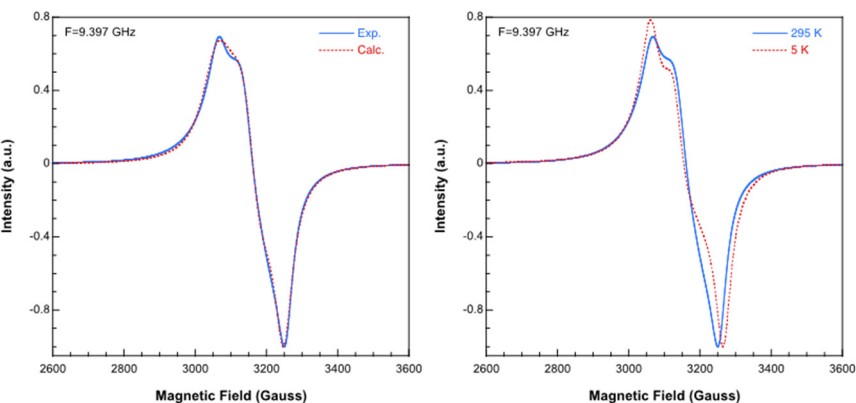

**Figure 6.** X-band ESR spectra of compound **1**. See text for the fitting parameters.

The Q-band (34.1 GHz) room temperature spectrum (Figure 7) supports this hypothesis. It shows the overlapping of two signals, a rhombic one with similar g values to those deduced from the X-band spectrum ($g_1$ = 2.192, $g_2$ = 2.136 and $g_3$ = 2.063; <g> = 2.130, G = 2.0) and an axial one with $g_{//}$ = 2.260 and $g_\perp$ = 2.063 (<g> = 2.130, G = 4.1). The g values of the axial signal are in good agreement with the structural characteristics of the $CuN_2O_3$ chromophore. Thus, it appears that the powder Q band spectrum shows the simultaneous presence of averaged and individual resonances for Cu1 and Cu1$^d$ ions. This behavior has already been observed previously, and occurs when the condition J>|g(Cu1)-g(Cu1$^d$)|βH is not fulfilled for all orientations of the magnetic field when the spectrum is recorded at Q-band [37]. Assuming near axial symmetry for the molecular g tensor, the following relations between the exchange and molecular components of the g tensors can be established [38]:

$$(g_1^{ex})^2 = g_\parallel^2 cos^2\alpha + g_\perp^2 sin^2\alpha$$

$$(g_2^{ex})^2 = g_\parallel^2 sin^2\alpha + g_\perp^2 cos^2\alpha$$

$$g_3^{ex} = g_\perp$$

$$cos2\alpha = \frac{g_1^{ex} - g_2^{ex}}{g_1^{ex} + g_2^{ex} - 2g_3^{ex}}$$

where $2\alpha$ is the canting angle between the parallel axis of the two tensors, i.e., the canting angle between the normals to the equatorial planes of the two interacting chromophores.

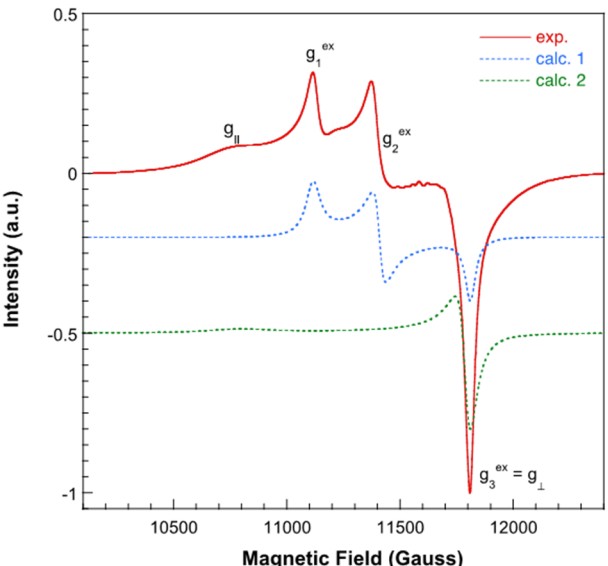

**Figure 7.** Q-band EPR spectrum recorded at room temperature. See text for the fitting parameters.

The value of $2\alpha$ that can be deduced from the last equation (73.9°) is practically the same as can be obtained from the crystallographic data (73.8°). Therefore, the molecular g values in this compound are: $g_{//} = 2.261$ and $g_{\perp} = 2.063$, in good agreement with a $d_{x^2-y^2}$ ground state, as expected for Cu(II) ions in elongated square-pyramidal geometry.

Moreover, the low value of the exchange interaction was confirmed by magnetic susceptibility measurements in the temperature range 2 to 300 K. The room temperature XmT value (0.838 cm³K/mol, $\mu_{eff}$ = 2.59 BM) is in good agreement with that expected for two isolated S = 1/2 ions with nearly quenched magnetic orbital contribution (g = 2.12). The magnetic effective moment remains practically constant down to 5 K, and slightly decreases below this temperature (Figure 8).

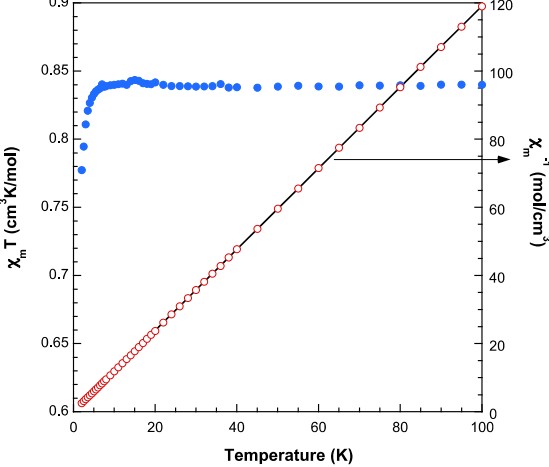

**Figure 8.** Magnetic behavior of compound X. The solid line corresponds to the Curie–Weiss fit (see text for the details).

The susceptibility data are well described by a Curie–Weiss expression for all the recorded temperature range, being Cm = 0.84 cm³·K·mol⁻¹ and $\theta$ = −0.1 K. The weakness of the magnetic interactions in this compound is not surprising, considering the long exchange pathways and the non-favorable disposition of the $d_{x^2-y^2}$ magnetic orbitals.

### 3.5. DFT Calculations

The theoretical study reported herein analyzes the supramolecular assemblies high-lighted above in Figures 3–5. Due to the polymeric nature of the compound, we used two different models of the system, depending on the assembly under investigation. In order to minimize the effect of dominant electrostatic forces, the here considered models were adjusted to be neutral, as detailed in Figure 9. For the evaluation of the OH···π interaction we used the mononuclear model, **A**, constructed using a half-molecule of EDTA and acetic acid as apical ligand. For the evaluation of the H-bonds and Him···Him π-stacking interactions, we used model B, that is dinuclear, the entire EDTA molecule is considered, and two water molecules are used as apical ligands. Model **A** was used for the theoretical study of the OH/π interaction because the water molecule also establishes an H-bonding interaction with carboxylate group in the X-ray structure. Model B was used to study the energetic features of the N–H···O interactions and π-stacking interactions, shown in Figure 5.

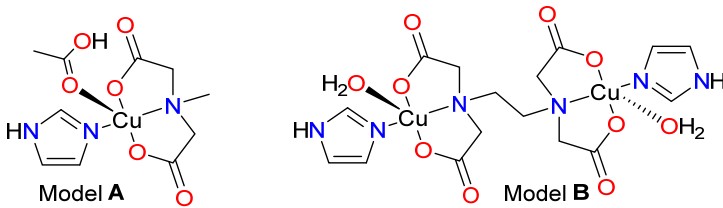

**Figure 9.** Representation of the theoretical models of **1** used to investigate the noncovalent interactions.

Figure 10a shows the MEP surface of model A, where the MEP maximum (+63 kcal/mol) is located at the NH group of Him ligand. Him coordination to Cu-atom increases the acidity of the NH group. The MEP minimum is located at the coordinated carboxylate group. Consequently, the NH···O interactions are electrostatically very favored. The MEP is also negative at the apical O-atom (−22 kcal/mol) and over the C5 atom of the Him ring (see Figure 2 for numbering scheme).

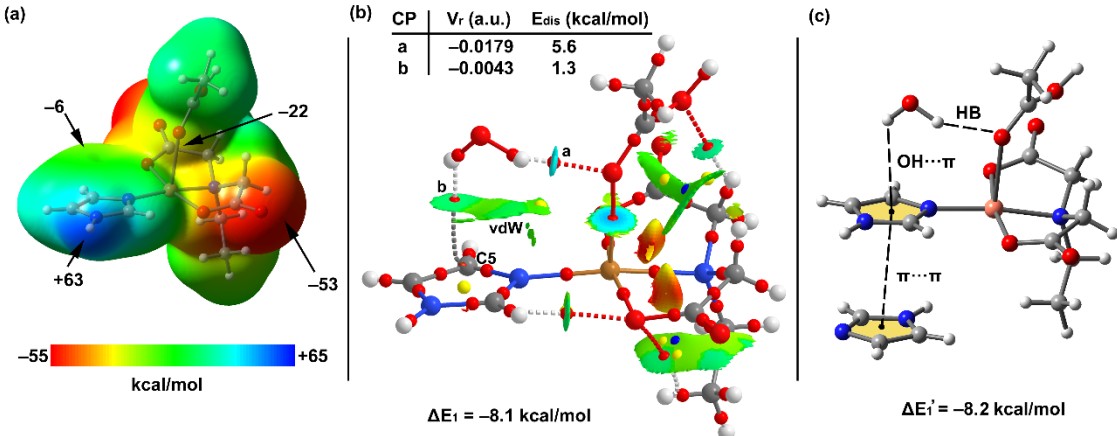

**Figure 10.** (**a**) MEP surface (isosurface 0.001 a.u.) of model **A** at the PBE0-D3/def2-TZVP level of theory. Values at selected points of the surfaces given in kcal/mol. (**b**) Combined QTAIM/NCIplot analysis of the complex between model A and water. Bond and ring critical points represented as red and ring spheres, respectively. For the NCIplot isosurface (0.5 a.u.), the $-0.04 < \text{sign}(\lambda_2)\rho < 0.04$ color scale was used. Gradient cut-off = 0.04 a.u. (**c**) Trimer used to evaluate the influence of the π-stacking on the OH···π interaction.

The Figure 10b shows the QTAIM distribution of critical points (CPs) and bond paths obtained for the complex of model A with water. It confirms the existence of the OH/π interaction that is characterized by a bond CP and bond path that connects the

H-atom of water to C5(Him). Moreover, it also shows the existence of the O–H⋯O H-bond characterized by the corresponding bond CP and bond path. The NCI index also confirms the existence of both interactions and shows that the H-bond (blue isosurface) is stronger than the C–H⋯π (green isosurface). It also shows that the green isosurface embraces the entire water molecule, thus suggesting a larger interaction region that embraces both H-atoms of water. We also evaluated the dissociation energy ($E_{dis}$) of each individual contact by using the methodology proposed by Espinosa et al. [39]. It is based on the potential energy density value measured at the bond CP that characterizes the H-bond. The CPs that characterize the contacts involving the water trimer are labelled as "**a**" and "**b**" in Figure 10b. The corresponding $V_r$ values, along with the $E_{dis}$ values derived from the formula ($E_{dis} = -0.5 \times V_r$), are indicated in the top-left corner of Figure 10b. It can be observed that the H-bond is stronger (5.6 kcal/mol) than the O–H⋯π interaction (1.3 kcal/mol), in agreement with the MEP surface analysis. It also agrees well with the color of the NCIplot isosurfaces for these HBs. The sum of the O–H⋯π and OH⋯O dissociation energies derived from the $V_r$ predictor is 6.9 kcal/mol, which is smaller than the interaction energy (in absolute value), $\Delta E_1 = -8.1$ kcal/mol. This difference can be attributed to the contribution of additional van der Waals interactions, characterized by the extended green NCIplot isosurface, labeled as vdW in Figure 10b.

As commented above, the imidazole ring establishes a π–π interaction at the opposite side of the OH⋯π interaction in the solid state (see Figures 3b and 5). Consequently, we further analyzed the influence of the π-stacking on the OH⋯π interaction, using the trimeric model shown in Figure 10c. The interaction energy (evaluate as a dimer, considering the π-π stacking as a monomer) is almost equivalent ($\Delta E_{1'} = -8.2$ kcal/mol) to $\Delta E_1$, thus evidencing that the interplay between the π-π and O–H⋯π interaction is negligible.

By using model B, the H-bonding and π-stacking interactions were analyzed, since they are also very relevant in the crystal packing. Figure 11a shows the H-bonded centrosymmetric homodimer where the NH group of each imidazole ligand interacts with the carboxylate group of the adjacent molecule, forming two symmetrically related and strong H-bonds. They are characterized by the corresponding bond CPs (denoted as "a"), bond paths and strong blue isosurfaces. The QTAIM analysis also revealed the formation of ancillary H-bonds established between the C–H bond of two imidazole ligands and the O-atom of the carboxylate groups (denoted as "b"). These C–H⋯O contacts are expected to be extremely weak since they are characterized by green isosurfaces and the H⋯O distance is very long (>3Å). The dimerization energy is very large ($\Delta E_2 = -37.4$ kcal/mol) due to the strong nature of the N–H⋯O H-bonds (9.2 kcal/mol each, as deduced using the $V_r$ energy predictor). Moreover, the $V_r$ value at the bond CP labelled as "b" (see Figure 8) confirms the weak nature of the C–H⋯O contact that is 0.3 kcal/mol.

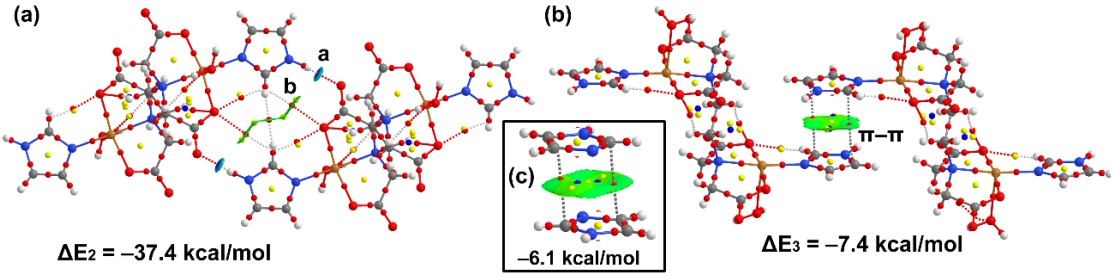

**Figure 11.** Combined QTAIM/NCIplot analysis of the H-bonded homodimer (**a**) and π–π homodimer (**b**) of compound **1**. (**c**) Combined QTAIM/NCIplot analysis of a π–π homodimer of two uncoordinated imidazole rings. Bond and ring critical points are represented as red and ring spheres, respectively. For the NCIplot isosurface (0.5 a.u.), the –0.04 < sign($\lambda_2$)ρ < 0.04 color scale was used. Gradient cut-off = 0.04 a.u. Only the intermolecular interactions have been represented in the NCIplot index analysis.

We also studied the antiparallel $\pi$-stacking interaction involving the $\pi$-clouds of the imidazole ligands, which are also relevant in the solid state of compound **1** (see Figure 11b). The dimerization energy is significant $\Delta E_3 = -7.4$ kcal/mol taking into consideration that it is a $\pi$-stacking interaction. The NCIplot index reveals the existence of a large isosurface located between the aromatic rings, and suggesting a large overlap of the $\pi$-systems. The QTAIM analysis shows two bond CPs and bond paths interconnecting the aromatic ligands. We studied the effect of the Cu-coordination on the interaction energy by computing a model where only the imidazole rings are considered. As a result, the interaction energy is reduced to $-6.1$ kcal/mol (see Figure 11c), thus evidencing that the coordination of the ligand to the Cu-atom reinforces the $\pi$-stacking, likely due to the polarization of the $\pi$-system that favors the antiparallel arrangement and increases the dipole$\cdots$dipole attraction.

## 4. Concluding Remarks

A new polymeric compound $\{[Cu_2(\mu_4\text{-EDTA})\,(Him)_2]\cdot 2H_2O\}_n$ was synthesized and X-ray characterized. Once again the use of basic carbonate of copper(II) proved the advantage of yielding $CO_2$, expected as the only by-product from its reaction with $H_4EDTA$ (besides water) [37]. The ESR spectra and magnetic measurements indicated that symmetry related Cu(II) centers connected by the bridging carboxylate groups behave magnetically not equivalently, enabling an exchange interaction larger than their individual Zeeman energies. The noncovalent interactions were analyzed in detail, revealing the formation of strong N–H$\cdots$O H-bonds and also antiparallel $\pi$-stacking interactions that are reinforced due to the coordination of the Him rings to the Cu-metal centers, as confirmed by DFT calculations. The role of the O–H$\cdots\pi$ interaction involving a lattice water molecule has been disclosed and characterized by QTAIM and NCI index computational tools. The energy associated to this interaction is weak compared to the strong H-bonds that dominate the X-ray packing. Interestingly the channels generated in the polymeric 3D-network confers to the here reported a host role, where the water molecules are distinctly guested. Finally, we underline that imidazole represents the five-membered ring moiety of all purine nucleobases and related natural or synthetic purine nucleosides. Thus, the O–H$\cdots\pi$ interaction here reported provides further support to a very recent investigation emphasizing such interactions between water and nucleobase in functional RNAs [40].

**Supplementary Materials:** The following are available online at https://www.mdpi.com/2073-4352/11/1/48/s1, Figure S1: Infrared spectrum of **1**; Figure S2: Electronic (Reflectance) spectrum of **1**; Figure S3: Termogravimetric analysis (TGA) of 1 with FT-IR spectra for identification of evolved gases.

**Author Contributions:** Conceptualization, J.N.-G. and M.E.G.-R.; methodology, all authors; software, J.M.G.-P., A.F. and A.C.; investigation, J.C.B.-S.; writing—original draft preparation, all authors, writing—review and editing, all authors; visualization, A.C., A.F. and J.N.-G. project administration, J.M.G.-P., J.N.-G.; funding acquisition, A.C., A.F. and J.N.-G. All authors have read and agreed to the published version of the manuscript.

**Funding:** MICINN of Spain (project PGC2018-102047-B-I00), MICIU/AEI of Spain (project CTQ2017-85821-R FEDER) and the Research groups FQM-283 and FQM-243 (Junta de Andalucía, Spain).

**Institutional Review Board Statement:** Not applicable.

**Informed Consent Statement:** Not applicable.

**Data Availability Statement:** Data is contained within the article or supplementary material.

**Acknowledgments:** We also thank all Projects for financial support. Also technical and human support provided by SGIker (UPV/EHU) is gratefully acknowledged.

**Conflicts of Interest:** The authors declare no conflict of interest.

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
