# Peer review of "H-Bonds, π-Stacking and (Water)O-H/π Interactions in (µ4-EDTA)Bis(Imidazole) Dicopper(II) Dihydrate"

_crystals, doi:10.3390/cryst11010048_

Round 1

Reviewer 1 Report

A new copper (II) EDTA imidazole complex has been synthesized and characterized. Its crystal structure was resolved by single crystal x-ray diffraction. ESR spectra and magnetic measurements indicated that symmetry related Cu(II) centers connected by the bridging carboxylate groups behave magnetically not equivalent. The noncovalent interactions in the solid state have been analyzed in detail, revealing short interatomic distances assigned to N–H···O Hydrogen bonding and π-stacking. DFT calculations were consistent with the observed results thus confirming the assumed attractive interactions.

The experimental work is properly described and the drawn conclusions are sound and consistent with the DFT models.

In the following some minor comments:

Why do the authors have chosen the described synthesis involving Malachite and not any other Cu(II) complex? The choice of the reactants should be commented in the text.

Lines 103-107: FT-IR spectrum and XRD analysis of the Cu compound have been made at different stages of the synthesis. Are the authors certain that both isolated crystals have the same crystallographic phase? Are FT-IR spectra and X ray powder spectra identical?

Lines 107-108: There is an error in the calculated values of the elemental analysis. Carbon does not account for 2.71 % but 32.71 (experimental: 30.67%) which is certainly a typing error. I agree that an elemental analysis confirms the purity of the compound based on the elemental formula. However, it is important in my opinion to rule out polymorphism (not rare phenomenon in molecular crystals) and the simplest way is to register X ray powder spectra in order to verify the homogeneity of the sample.

The manuscript suffers from grammatical errors. Some examples in the introduction part of the paper:

Line 34: form up (instead of “forms up”)

Line 36: compounds (instead of “compound”)

Line 37: in using the / in the use of the (instead of “in use the”

Line 45: has not (instead of “have not”)

Line 46: this kind of compounds has been…

Line 48: sentence to be revised

Lines 51-53: sentence to be revised

Author Response

First, we would like to thank this reviewer for her/his careful reading of the manuscript, corrections and interesting suggestions. We have taken into consideration his/her corrections and revised the manuscript accordingly as detailed below. The changes made have been highlighted using a yellow background whenever possible.

Comment 1: Why do the authors have chosen the described synthesis involving Malachite and not any other Cu(II) complex? The choice of the reactants should be commented in the text.

Answer 1: Since 1985 we have used this strategy successfully. Moreover, we are pleased to add in Concluding remarks the following sentence: ‘Once again the use of basic carbonate of copper(II) has proved the advantage to yield CO2, expected as only by-product from its reaction with H4EDTA (besides water) [29].’

Comment 2: Lines 103-107: FT-IR spectrum and XRD analysis of the Cu compound have been made at different stages of the synthesis. Are the authors certain that both isolated crystals have the same crystallographic phase? Are FT-IR spectra and X ray powder spectra identical?

Answer 2: Yes, we are! Not only because the shape of crystals during sample collection is the same, but also the IR spectra of all samples are identical.

Comment 3: Lines 107-108: There is an error in the calculated values of the elemental analysis. Carbon does not account for 2.71 % but 32.71 (experimental: 30.67%) which is certainly a typing error.

Answer 3: Thanks for your comment! The revised data of elemental analysis is written as:

Elemental analysis (%): Calc. for C16H24Cu2N6O10: C 32.71, H 4.12, N 14.31, Cu (as CuO) 27.08; Found: C 32.67, H 4.10, N 14.29, Cu 27.09 (as CuO, final residue at 950 °C, in the TGA curve).

Comment 4: I agree that an elemental analysis confirms the purity of the compound based on the elemental formula. However, it is important in my opinion to rule out polymorphism (not rare phenomenon in molecular crystals) and the simplest way is to register X ray powder spectra in order to verify the homogeneity of the sample.

Answer 4: We totally agree. Indeed we are working in crystallography since 1985… And we use high resolution powder X-ray diffraction when seems to be necessary. In practice, nowadays several single crystals of the studied compound are mounted in the diffractometer, ruling out the existence of polymorphism in the compound reported here.

Comment 5: The manuscript suffers from grammatical errors. Some examples in the introduction part of the paper:

Line 34: form up (instead of “forms up”).

Line 36: compounds (instead of “compound”).

Line 37: in using the / in the use of the (instead of “in use the”).

Line 45: has not (instead of “have not”).

Line 46: this kind of compounds has been…

Line 48: sentence to be revised.

Lines 51-53: sentence to be revised. 

Answer 4: We have revised and addressed these mistakes. Thank you!

Reviewer 2 Report

In he manuscript entitled "H-bonds, π-Stacking and (water)O-H/π Interactions in (µ4-EDTA)bis(Imidazole)Dicopper(II) Dihydrate" the authors have described the synthesis, characterization and AIM analysis of one new polymeric crystalline compound.

The results are sound, but I have some questions and/or comments:

-The DFT calculations are single point energy calculations of the crystal structure or optimizations. Please specify.

-Why the authors have selected PBE0 hybrid functional over other pure DFT functionals or range separated orbitals that are sometimes used to analyze solids. Please clarify.

-In the synthesis of the crystal the authors have used an open Kitasato flask to perform the reaction. From my point of view is kind of inusual. Is it mandatory to use a Kitasato flask for the reaction to happen? 

-Do Figure 3a and Figure 5 represent the same X-ray structure? Please delete one of them to avoid repetition

-In Figures 3b and Figure 4 truncated structures are represented, please indicate in the Figure caption.

-Concerning the DFT calculations, it sounds strange that the reported OH-pi interaction is stronger than the pi-pi imidazole interaction. Have the authors found any similar example in the literature?. On the other hand, have the authors considered that a central imidazole ring interacts with water in one side and with another imidazole ring in the other  side? Does it affect the interaction energies? 

In page 12 there are several minor typos:

lines 321 and 327: Figure 7b should Figure 10b 

line 329 figure 8(a) should be Figure 11(a)

caption of Figure 8 should be Figure 11 and should be completed to indicate what (b) and (c) stands for. 

Author Response

First, we would like to thank this reviewer for her/his careful reading of the manuscript, corrections and interesting suggestions. We have taken into consideration his/her corrections and revised the manuscript accordingly, as detailed below. The changes made have been highlighted using a yellow background whenever possible.

Comment 1: The DFT calculations are single point energy calculations of the crystal structure or optimizations. Please specify.

Answer 1: Yes, they are. It has been clarified in the computational methods

Comment 2: Why the authors have selected PBE0 hybrid functional over other pure DFT functionals or range separated orbitals that are sometimes used to analyze solids. Please clarify.

Answer 2: Thank you for this comment. It has been clarified in the computational methods

Comment 3: In the synthesis of the crystal the authors have used an open Kitasato flask to perform the reaction. From my point of view is kind of inusual. Is it mandatory to use a Kitasato flask for the reaction to happen? 

Answer 3:  In this case it is convenient to use a Kitasato flask instead of a more common glassware (Erlenmeyer) because the synthesis produces CO2 (as main by-product). The use of a Kitasato flask allows us to cover the upper mouth (to avoid projections when stirring and heating the malachite digestion) and at the same time to maintain an ‘open system’ through the hose barb (to avoid overpressure).

Comment 4: Do Figure 3a and Figure 5 represent the same X-ray structure? Please delete one of them to avoid repetition.

Answer 4: We prefer to keep both figures for the following reason. Please note that Figure 3 emphasizes the intermolecular interactions that contribute to the stability of the channels in the crystal, whereas Figure 5 represents the host function of the polymer where disordered water molecules are guested as well as the O-H/pi interaction built by one of two position refined for water.

Comment 5: In Figures 3b and Figure 4 truncated structures are represented, please indicate in the Figure caption.

Answer 5: The caption of this Figures are rewritten according to the comment of this reviewer.

Comment 6: Concerning the DFT calculations, it sounds strange that the reported OH-pi interaction is stronger than the pi-pi imidazole interaction. Have the authors found any similar example in the literature? On the other hand, have the authors considered that a central imidazole ring interacts with water in one side and with another imidazole ring in the other side? Does it affect the interaction energies? 

Answer 6: This is a very interesting comment and suggestion. We have further analyzed the effect of including a second imidazole ring in the opposite side. The new results have been commented in the manuscript, although we have found that the influence is very small. Regarding the first question, the OH···π interaction is in fact weaker, as revealed by the QTAIM Analysis (see Figure 10b, top). The OH···π assembly is stronger than the π-π stacking of imidazole rings due to the ancillary H-bond (O–H···O) between the water molecule and the coordinated carboxylate group.

Comment 7: In page 12 there are several minor typos:

lines 321 and 327: Figure 7b should Figure 10b 

line 329 figure 8(a) should be Figure 11(a)

caption of Figure 8 should be Figure 11 and should be completed to indicate what (b) and (c) stands for. 

Answer 7: Thanks again! These typos are removed in the text.

Reviewer 3 Report

This work is based on the synthesis, characterization and theoretical studies of one copper complex. The X-ray crystal data is being carefully detailed and contrasted with DFT calculations.

In my opinion, the paper deserves its publication in this journal after some minor corrections:

  • check the affiliations of the authors in the SI. They must be corrected as shown in the main article.
  • the IR spectra presented in SI are very bad quality. It seems that they are very diluted with a large CO2 peak at 2200 cm-1 that masks the rests of the peaks that are shown in a very expanded vision of the spectra. Nevertheless, the peaks are very weak and they are difficult to believe that correspond to those assigned in the experimental section.
  • English must be checked. Some examples: in the introduction, "among all this possibilities" (must be "these possibilities) or section 2.4 "about week at r.t.)" must be said "about one week at r.t.".
  • Section 2.4. The authors say that the reaction was performed in a kitasato under stirring and heating and I hope they would like to say in a glass flask not a kitasato. A kitasato is used only for filtration and not stirring neither heating is there performed.
  • Elemental analysis: C data must be checked. The theoretical value (2.71) is much different from the experimental one (30.67). It must be an error.
  • Vis-UV must be changed as UV-vis and the ~15000cm-1 value must be removed since experimentally we speak in nm and it does not make any sense to introduce the corresponding value in cm-1 and even less in an approximation number.

Round 2

Reviewer 2 Report

The revised version of the manuscript has been improved and all the comments were successfully addressed.

I have only one minor comment:

-In the theoretical methods, the authors included references 20-27 to explain why they perform the calculations at PBE0/def2-TZVP level, however references 20-24 deals with halogen bonding interactions (not present in the studied system) published by the authors. 

Besides that, in my opinion, the article is suitable to be published in Crystals.

Author Response

First, we would like to thank this reviewer for her/his careful second reading of the manuscript. We have taken into consideration his/her additional comment and revised the manuscript accordingly as detailed below. The changes made have been highlighted using a yellow background.

Comment 1: In the theoretical methods, the authors included references 20-27 to explain why they perform the calculations at PBE0/def2-TZVP level, however references 20-24 deals with halogen bonding interactions (not present in the studied system) published by the authors.

Answer 1: Thank you for catching this. The sentence in the theoretical methods has been modified as follows:

This level of theory (functional and basis set) has been used before to study noncovalent interactions in the solid state [20-24], including those analyzed in this work [25-27].